# Effects of Mixed Adding Crude Extracts of β-Glucosidases from Three Different Non-*Saccharomyces* Yeast Strains on the Quality of Cabernet Sauvignon Wines

**DOI:** 10.3390/jof8070710

**Published:** 2022-07-04

**Authors:** Jing Liao, Shuangmei Zhang, Xiuyan Zhang

**Affiliations:** College of Food Science and Technology, Huazhong Agricultural University, Wuhan 430070, China; jingliaohzau98@163.com (J.L.); zhangsm504@163.com (S.Z.)

**Keywords:** β-glucosidases, non-*Saccharomyces* yeasts, mix adding, wine aroma

## Abstract

The aim of this study was to investigate the effects of crude extracts of β-glucosidase from *Issatchenkia terricola* SLY-4, *Pichia kudriavzevii* F2-24 and *Metschnikowia pulcherrima* HX-13 (termed as SLY-4E, F2-24E and HX-13E) on the flavor complexity and typicality of Cabernet Sauvignon wines. The grape must was fermented using *Saccharomyces cerevisiae* with single or mixed SLY-4E, F2-24E and HX-13E. The physicochemical characteristics, volatile aroma compounds, total anthocyanins and sensory attributes of the wines were determined. Adding SLY-4E, F2-24E and HX-13E in wines resulted in a decrease in the anthocyanin content, total acids and volatile acids in wines but an increase in the content of terpenes, benzene derivatives, higher alcohols and esters, which may enhance wine sensory qualities and result in loss of wine color. Different adding strategies of β-glucosidase led to a variety of effects on wine aroma. S/H/F-Ew significantly increased the content of benzene derivatives, higher alcohols and long-chain fatty acid esters, which enhanced the fruity and floral flavor of wines. F2-24E significantly increased the content of short- and medium-chain fatty acid esters, acetate esters and carbonyl compounds. The results indicated that the mixed addition of non-*Saccharomyces* crude extracts and co-fermentation with *S. cerevisiae* could further improve wine flavor quality.

## 1. Introduction

Wine is popular among consumers for its nutritional and healthy properties. Fermentation with *Saccharomyces cerevisiae* alone is a common method for commercial wine production to ensure its uniform flavor quality but usually causes poor flavor complexity and typicality of wine [1], resulting in the reduction in competitiveness. Under this context, improving the flavor complexity and typicality of wine has become the common goal pursued by scientists. It was reported that β-glucosidases could hydrolyze nonvolatile flavor precursors and generate aroma substances with volatile flavor [2,3]. Therefore, co-fermenting with non-*Saccharomyces* and *S. cerevisiae* yeasts [4,5,6,7], or adding crude extracts or purified enzymes from non-*Saccharomyces* yeasts before fermentation [8,9,10] could improve the flavor complexity and typicality of wines. Moreover, different non-*Saccharomyces* yeast strains or their crude extracts of enzyme might produce different aroma compound profiles in wines [10]. Previous research indicated that *Issatchenkia terricola* SLY-4, *Pichia kudriavzevii* F2-24 and *Metschnikowia pulcherrima* HX-13 with *β*-glucosidases activities or their crude extracts of β-glucosidase (termed as SLY-4E, F2-24E and HX-13E) could improve the flavor complexity and typicality of wines and present different aroma compound profiles by co-fermenting with *S. cerevisiae* [10,11]. However, it is still unclear whether adding a combination of SLY-4E, F2-24E and HX-13E into must can further improve the flavor complexity and typicality of wines.

The aim of this study was to investigate the effect of adding mixed SLY-4E, F2-24E and HX-13E crude extracts of β-glucosidase on the flavor complexity and typicality of Cabernet Sauvignon wines. Wines were fermented by *S. cerevisiae* with the addition of SLY-4E, F2-24E and HX-13E. The physicochemical characteristics, aroma components and sensory characteristics of wines were studied.

## 2. Materials and Methods

### 2.1. Strains and Medium

*Issatchenkia terricola* SLY-4 (named as SLY-4), *Pichia kudriavzevii* F2-24 (named as F2-24) and *Metschnikowia pulcherrima* HX-13 (named as HX-13) were isolated from Helan Mountain vineyards in China. *Saccharomyces cerevisiae* used in this study is a commercial strain Actiflore^®^ F33(Laffort, Bordeaux, France).

Fermentation medium (10 g/L yeast extract, 20 g/L peptone, 20 g/L glucose, 3 g/L NH_4_NO_3_, 4 g/L KH_2_PO_4_, 0.5 g/L MgSO_4_·7H_2_O and 10 mL/L Tween 80) was used to ferment β-glucosidases with non-*Saccharomyces* yeasts.

### 2.2. Preparing Crude Extracts of β-Glucosidase

The starter culture of SLY-4, F2-24 and HX-13 were inoculated into the fermentation medium at 10^6^ CFU/mL and incubated at 28 °C for 72 h. The fermentation broth was centrifuged (8500 rpm, 5 min) at 4 °C. The sediment was collected, washed with disodium hydrogen phosphate-citric acid buffer (P-C buffer, containing 10.2 g/L citric acid monohydrate and 36.8 g/L Na_2_HPO_4_·12H_2_O in deionized water, pH 5.0) and centrifuged (Neofugel15R, Heal Force, Shanghai, China). A total of 5 g of yeast cell precipitate was dissolved in 20 mL of P-C buffer and homogenized using a high-pressure homogenizer (Scientz-150, Scientz, Ningbo, China) at 100 MPa three times. The cell crushing solution was centrifuged (8000 rpm, 15 min) at 4 °C and the supernatant was collected [2]. The SLY-4E, HX-13E and F2-24E were precipitated by adding different-saturation (NH4)_2_SO_4_ solution (20%, 30%, 40%, 50%, 60%, 70%, and 80%) into the supernatant of cell crushing solution and placed in an ice bath for 2 h. Then, the salting-out crude β-glucosidase was dissolved in 20 mL of 0.05 mol/L Tris-HCl buffer (pH 7.4) and desalted by dialysis with the same buffer. The enzyme solutions were concentrated by PEG 20,000 (Solarbio, Beijing, China).

### 2.3. Analyzing the β-Glucosidase Activity

The β-glucosidase activity was analyzed according to the method described by Zhang et al. (2020) [10] with modifications. A total of 0.1 mL of crude extracts of β-glucosidase and 0.2 mL of 5 mmol/L *p*-nitrophenyl β-d-glucopyranoside (*p*-NPG) in phosphate buffer were mixed and incubated at 50 °C for 30 min; then, the reaction was inactivated by 2.0 mL 1.0 mol/L Na_2_CO_3_ solution. The absorption value of the reaction solution was measured at 400 nm (Multiskan GO, Thermo Scientific, Waltham, MA, USA). The distilled water (200 μL) was added to replace p-NPG in control. One unit activity of β-glucosidase (U) was defined as the quantity of enzyme required to produce 1.0 μmol *p*-nitrophenol per minute under the reaction condition described.

### 2.4. Stability of β-Glucosidase in Crude Extracts

The SLY-4E, HX-13E and F2-24E extracts were added into modified P-C buffer (pH 4.0, containing 23.7% glucose and 12% ethanol) and stored at 20 °C for 14 days, respectively. The *β*-glucosidase activity was determined every 2 days.
$$\mathrm{Relative}\;\mathrm{activity}\;\left(\%\right)=\frac{\mathrm{Final}\;\beta \text{-}\mathrm{glucosidase}\;\mathrm{activity}}{\mathrm{Initial}\;\beta \text{-}\mathrm{glucosidase}\;\mathrm{activity}}\times 100\%$$

### 2.5. Wine Fermentation

Must (237.1 g/L sugar calculated as glucose and 5.3 g/L acid calculated as tartaric acid) was incubated at 80 °C for 45 min. Then, 200 mL sterilized must containing 60 mg/L total SO_2_ was macerated at 4 °C for 12 h in a 250 mL glass bottle. A total of 1U of SLY-4E, HX-13E and F2-24E extract was added to the must according to fermentation strategies (Table 1). Fermentation was carried out at 20 °C by inoculating 10^6^ CFU/mL *S. cerevisiae*. Fermentation was considered and finished when the residual sugar was below 4 g/L.

### 2.6. Analysis of Wines

The content of alcohol, total acids and volatile acids was determined through methods recommended by the International Organization of Vine and Wine (OIV, 2021).

The anthocyanins content in wines was determined according to the methods described by Jungmin Lee et al. (2007) [12]. After centrifugation (8000 rpm, 20 min) of the wines (10 mL), the supernatant (1 mL) was collected and diluted appropriately with KCl buffer (0.025 mol/L, pH 1.0) and CH_3_COONa buffer (0.4 mol/L, pH 4.5), respectively. Then, the absorbance value of wines was determined at 520 nm and 700 nm after standing for 15 min in the dark at room temperature and distilled water was used as control. The anthocyanin content was expressed as cyanidin glucoside and calculated as follows:$$\mathrm{Anthocyanin}\;\mathrm{content}\;\left(\mathrm{mg}/L\right)=\frac{\left[\left(\mathrm{OD}520-\mathrm{OD}700\right)\mathrm{pH}1.0-\left(\mathrm{OD}520-\mathrm{OD}700\right)\mathrm{pH}4.5\right]\times M\times \mathrm{DF}\times 1000}{\varepsilon \times L}$$

M represents the molecular weight of cyanidin glucoside (449.4 g/mol). DF represents the dilution rate of wines. ε represents the molar extinction coefficient for cyanidin glucoside (26,900 L/mol.cm). L represents the width of the cuvette (1 cm).

The volatile compounds were analyzed by headspace solid-phase micro-extraction according to Shi et al. (2019) [11]. Volatile compounds were extracted by 50/30 µm DVB/CAR/PDMS fiber (Supelco, Bellefonte, PA, USA) and analyzed by gas chromatograph (Agilent 6890N, Agilent, CA, USA) with a DB-5 capillary column (30 m × 0.32 mm × 0.25 µm) coupled to a mass spectrometer (Agilent 5975B, Agilent, CA, USA). Wines (8 mL), NaCl (2 g) and cyclohexanone (40 µg/L, 100 µL) as internal standard were added into a headspace bottle (20 mL) and bathed at 40 °C for 15 min. The fiber was pushed into the headspace for 30 min and immediately desorbed in the injector of gas chromatography at 250 °C for 5 min. Temperature increasing conditions of GC analysis were as follows: 40 °C to 130 °C at 3 °C/min, and then to 250 °C at 4 °C/min. The injector and detector temperatures were set at 250 °C and 260 °C, respectively. The mass spectrometer was operated in electron impact ionization mode at 70 eV, and ion source temperature was 250 °C. Volatile compounds were identified by comparing the MS fragmentation pattern of each compound with that in database Wiley 7.0 and NIST05. The following formula was used for the calculation of the compound content:$$\mathrm{Compound}\;\mathrm{content}\;\left(µg/\mathrm{mL}\right)=\frac{\mathrm{GC}\;\mathrm{peak}\;\mathrm{areas}\;\mathrm{of}\;\mathrm{the}\;\mathrm{compound}\times \mathrm{Quality}\;\mathrm{of}\;\mathrm{cyclohexanone}\;\left(µg\right)}{\mathrm{GC}\;\mathrm{peak}\;\mathrm{area}\;\mathrm{of}\;\mathrm{cyclohexanone}\times \mathrm{Volume}\;\mathrm{of}\;\mathrm{wine}\;\mathrm{sample}\;\left(\mathrm{mL}\right)}$$

The sensory quality of wines was analyzed according to the methods described by Shi et al. (2019) [11] with little modification.

### 2.7. Data Analyses

Data statistics and graphs were performed by Graphpad prism 6.0. One-way analysis of variance (ANOVA) was conducted by SPSS 19.0 (SPSS Inc., Chicago, IL, USA). Principal component analysis (PCA) was performed with SIMCA-P 14.1 (Umetrics AB, Umea, Sweden). Heatmap visualization analysis of the wines and their flavor compounds was performed with TBtools_JRE1.6 after the Z-score standardization.

## 3. Results and Discussions

### 3.1. Stability of Crude Extracts of β-Glucosidase

The stability of crude extracts of β-glucosidase was analyzed. The relative activity of SLY-4E, HX-13E and F2-24E was 65.79%, 75.97% and 64.93%, respectively, after 14 days of incubation in modified P-C buffer (Figure 1). The relative activity of SLY-4E, HX-13E and F2-24E was higher than that of crude extract of β-glucosidase from *Rhodotorula mucilaginosa*, with nearly 63% relative activity when it was incubated for 14 days under 20% glucose and 10% ethanol [2], and that of pure β-glucosidase from *Pichia guilliermondii* G1.2, with 40% relative activity when it was incubated under 1000 mM (18%) glucose [13].

In fact, the activity of β-glucosidase might be affected by the concentration of glucose or ethanol [14,15]. It was increased at low ethanol concentration by increasing the glycosyl transferase activity but it was decreased at high concentration of glucose (10%) or ethanol (≥10% *v*/*v*) by altering its protein structure [16,17,18,19]. These results indicated the activity of SLY-4E, HX-13E and F2-24E was affected by a high concentration of glucose and ethanol but was relatively stable during wine fermentation.

### 3.2. Physicochemical Characteristics of Wines

Physicochemical characteristics showed that the content of alcohol (12.07–12.56%) and total acids (5.20–5.81 g/L) were not significantly different among the wines (Table 2). Compared with SCw (0.68 g/L), the volatile acids in wines (0.49–0.65 g/L) with added β-glucosidase, except H-Ew (0.68 g/L), decreased, which had no relation with the adding strategies of β-glucosidase (Table 2). The volatile acid content of wines also decreased after being fermented by yeast strains with β-glucosidase activity [20,21]. Excessive volatile acids (>1.2 g/L) would bring undesirable flavors to wine [22]. These results indicate that adding SLY-4E, F2-24E and HX-13 into must would improve flavors by decreasing the volatile acid content. The reason for the effect of crude extracts of β-glucosidase on the volatile acid content in wines is still unclear. Further, the addition of SLY-4E, F2-24E and HX-13E in wines could also decrease the content of anthocyanin (163.30–184.96 mg/L) (Table 2). In particular, it shows that the anthocyanin content of H-Ew was 163.30 mg/L, which was lower than S-Ew and F-Ew. However, the anthocyanin contents of S/H-Ew (173.08 mg/L) and H/F-Ew (172.17 mg/L) and S/H/F-Ew (184.96 mg/L) were higher than H-Ew, which showed that the combinational addition of SLY-4E, F2-24E and HX-13E could attenuate the declining trend. Consistent with the result of Vernocchi et al. (2011) [21], the wines fermented by *S. cerevisiae* with higher β-glucosidase activity had lower anthocyanins concentrations. β-glucosidases can hydrolyze the glycosidic bond of anthocyanin to release free anthocyanidins, and free anthocyanidins easily degraded into colorless compounds, which would have a negative effect on the color of wine [23,24]. The results showed that adding crude extracts and purified β-glucosidase from yeasts could decrease the anthocyanin content of wines, and those from different yeasts would exhibit different hydrolyzing activity of anthocyanins.

### 3.3. Varietal Aroma Compounds of Wines

Eight varietal aroma compounds were detected in the wines. Compared with SCw (1.14 mg/L), the content of varietal aroma compounds in wines with added SLY-4E, F2-24E and HX-13E (1.26–1.86 mg/L) was significantly higher (Table 3). The content of varietal aroma compounds in S/H/F-Ew was higher than S/H-Ew, S/F-Ew and H/F-Ew, which was higher than SLY-4E, F2-24E, and HX-13E. All the wines with added F2-24E had a higher content of varietal aroma compounds than other wines with SLY-4E or HX-13E. González-Pombo et al. (2011) [8] also reported that the purified β-glucosidase from *I. terricola* could significantly increase the content of varietal aroma compounds in white Muscat wine. It reflected that adding β-glucosidase could increase the varietal aroma content of wines, which usually exist as nonvolatile odorless glycosides, and β-glucosidase could hydrolyze the aroma compounds to release volatile aroma compounds and enhance the content of varietal aroma compounds [20,25,26,27]. In addition, the content of volatile aroma compounds and the number of β-glucosidase combined added in wines are directly proportional. Therefore, it is likely that there was a synergistic effect between SLY-4E, F2-24E and HX-13E, which performed better in releasing varietal aroma compounds. Zietsman et al. (2010) [28] found a synergistic effect of both α-l-arabinofuranosidase and β-glucosidase towards diglycosidically bonded monoterpenes, which could enhance the varietal aroma compound content. However, there are no reports about the synergistic effect of β-glucosidase in increasing the content of varietal aroma compounds. As the PCA plot shows (Figure 2), 79.8% variance was explained by eight varietal aroma compounds, and PC1 and PC2 accounted for 65.2% and 14.6% variance, respectively. S/F-Ew, H/F-Ew and S/H/F-Ew grouped with 1-hexanol. S/H/F-Ew grouped with citronellol, 1-octen-3-ol and geraniol. S/F-Ew grouped with geranyl acetone, while other wines did not group with any of the varietal aroma compounds. Among the varietal aroma compounds, the odor active values (OAV) of linalool and 1-octen-3-ol were above 1.0, which would present floral and fruity flavor. These results indicate that adding mixed SLY-4E, F2-24E and HX-13E crude extract could effectively promote the releasing of all the detected varietal aroma compounds. Adding S/F-E could promote the releasing of geranyl acetone. Using various β-glucosidase to release varietal aroma compounds had a synergistic effect on releasing varietal aroma compounds, especially 1-hexanol. There are no reports about the synergistic effect of β-glucosidase in releasing different aroma compounds. Systematic studies on the mechanism of the synergistic effect of β-glucosidases in increasing the varietal aroma compound should be investigated in the future.

### 3.4. Fermentative Aroma Compounds

The content of fermentative aroma compounds in S-Ew, H-Ew, F-Ew, S/H-Ew, S/F-Ew, H/F-Ew and S/H/F-Ew (338.21–514.14 mg/L) was significantly higher than in SCw (221.66 mg/L). S/H/F-Ew contained the highest aroma compounds (514.14 mg/L) (Table 3). Among the fermentative aroma compounds, the content of benzene derivatives (93.70–136.26 181 mg/L), higher alcohols (188.42–336.42 mg/L) and esters (29.84–48.31 mg/L) in wines with added crude extract of β-glucosidase were higher, compared with those of SCw, but the content of fatty acids and carbonyl compounds showed no significant difference (Figure 3).

Although many studies have reported that the addition of β-glucosidase impacted the fermentation aroma, β-glucosidase from different strains affected the fermentation aroma differently [29]. It was shown that β-glucosidase from *H. uvarum* and *Candida easanensis* JK8 increased the contents of benzene derivatives in wines [30,31]. Phenyl ethanol was the most abundant compound in five benzene derivatives, which is one of the aglycons of aroma precursors in grapes and could be released by β-glucosidase [32,33]. Moreover, phenyl ethanol was the synthetic material for phenylethyl acetate and phenylacetaldehyde generated from phenylalanine by the Strecker degradation pathway, which may be affected by amino acid content and composition in musts [34]. Higher alcohols play a crucial role in wine aroma and could enhance floral and fruity flavor in wines [35]. It was reported that β-glucosidase from *H. uvarum* and *Kloeckera apiculata* increased the higher alcohol content of wine, especially isoamyl alcohol [2,36]. However, according to Zhang et al. (2020) [10], adding SLY-4E, F2-24E and HX-13E decreased the higher alcohol content. These differences could be attributed to the grape must used in this study containing a high amount of nitrogen. It was confirmed that amino acids could be transformed into higher alcohols via the Ehrlich pathway, and increased nitrogen in musts can result in higher alcohol contents in wines [37]. Esters present fruity and floral characteristics to wine, contributing to the presence of wine flavor [38]. β-glucosidase from different origins resulted in various levels of increase in the content of esters, showing high strain-dependent variations. Crude β-glucosidase from *Pichia fermentans* and *H. uvarum* increased the content of ethyl octanoate, ethyl decanoate and ethyl laurate, which are medium-chain fatty acids esters [9,30]. On the other hand, β-glucosidase from *Pichia anomala* could increase the concentrations of ethyl hexanoate and ethyl decanoate [39]. It is possible that the substrate specificity of β-glucosidase varies from the different yeast strains and became more suitable for substrate hydrolysis in the ester formation biosynthetic pathways [40]. These results showed that adding β-glucosidases to wines could change the fermentative aroma profiles by increasing the content of benzene derivatives, higher alcohols and esters.

Wines were classified into five groups (Figure 4), which include (i) SCw, (ii) F-Ew, (iii) H/F-Ew, S/H-Ew and S/F-Ew, (iv) S/H/F-Ew, and (v) S-Ew and H-Ew. Hierarchical cluster analysis (HCA) showed that all the detected fermentative aroma compounds were classed into I, II, III and IV. SCw was abundant in Class I, which contained nonanal, 2-methylbutyric acid and isopentanoic acid and presented herb aroma. S-Ew and H-Ew were abundant in Class I and Class IV, which contained phenylacetaldehyde, ethyl heptanoate, 2-methyl-1-butanol and carbonyl compounds (2,3-pentanedione, decanal and 2,4-dimethoxybenzaldehyde). F-Ew was abundant in Class III and IV, including short- and medium-chain fatty acid esters, acetate esters and carbonyl compounds. In Class III, the OVA of isoamyl acetate, ethyl butyrate, ethyl hexanoate and ethyl octanoate was greater than 1, which improved fruity aroma in wines. H/F-Ew, S/H-Ew and S/F-Ew and S/H/F-Ew were higher in Class II, including benzene derivatives (except phenylacetaldehyde), higher alcohols (except 2-methyl-1-butanol), ethyl sebacate, long-chain fatty acid esters (ethyl laurate, ethyl tetradecanoate and hexadecanoic acid, and ethyl ester), and other esters and fatty acids. In Class II, the OVA of phenyl ethanol, phenethyl acetate, isoamyl alcohol, 3-methyl-1-pentanol, 1-nonanol, ethyl decanoate, ethyl laurate and isoamyl octanoate were greater than 1. These compounds presented fruity, fatty, floral and earthy notes and improved the complexity of aroma in H/F-Ew, S/H-Ew, S/F-Ew and S/H/F-Ew. This indicated that F2-24E was more favorable than SLY-4E and HX-13E in releasing short- and medium-chain fatty acid esters, acetate esters and carbonyl compounds when added alone. Adding mixed SLY-4E, F2-24E and HX-13E crude extract mainly improved the production of benzene derivatives, esters and higher alcohols, and S/H/F-Ew had the strongest ability to release these aroma substances. This reflects that mixed addition of SLY-4E, F2-24E and HX-13E affected the increase in acetate esters, short- and medium-chain fatty acids ester. However, there are no studies reporting the effect of mixed addition of β-glucosidase in releasing fermentative aroma compounds. These different aroma profiles of wines are probably caused by the other components in SLY-4E, F2-24E and HX-13E crude extract which affect the must composition of wines and need to further be determined after purified SLY-4E, F2-24E and HX-13E.

### 3.5. Sensory Analysis

Finally, a sensory analysis was carried out to evaluate the quality attributes of the wines, including flavor, aroma and color, which have effects on consumer acceptance (Figure 5). Samples S-Ew, H-Ew, F-Ew, S/F-Ew, H/F-Ew, S/F-Ew and S/H/F-Ew had higher scores in floral (7.00, 7.11–8.22), fruity (7.22, 7.76–8.50), taste (6.00, 6.00–8.67) and acceptance (6.17, 6.33–8.50) but lower scores in appearance (9.17, 8.33–8.83) compared to SCw. Ma et al., (2017) reported that crude β-glucosidases from *P. fermentans* could promote the release of varietal aroma compounds and fortified fruity and floral traits [9]. Taste qualities in wine mainly contained sweetness, sourness and bitterness, contributed by sugars, organic acids and ethanol, respectively. Acetic acid is mostly responsible for the sour and vinegary smell and taste in wines [41,42]. We analyzed the volatile acid of wines, which mainly contained acetic acid. Therefore, adding SLY-4E, F2-24E and HX-13E to wines decreased the content of volatile acids and improved wine taste. In addition, there were other compounds that imparted better taste to S/H/F-Ew and this should be investigated in the future. The lower appearances score in wines with added SLY-4E, F2-24E and HX-13E was probably due to the decreased anthocyanin content [43]. S/F-Ew and S/H/F-Ew had higher appearance scores and anthocyanin content than other adding strategies, which demonstrated that S/F-E and S/H/F-E were more favorable for wine color development. Wines’ quality attributes include flavor, aroma and color, which impact consumer acceptance. Results showed that adding SLY-4E, F2-24E and HX-13E could improve wine sensory qualities.

## 4. Conclusions

In this study, three crude extracts of β-glucosidases from non-*Saccharomyces* yeasts SLY-4 (*I. terricola*), F2-24 (*P. kudriavzevii*) and HX-13 (*M. pulcherrima*) were evaluated under simulated winemaking conditions. Adding SLY-4E, F2-24E and HX-13E to wines decreased the content of total acids and volatile acids, and exhibited different hydrolyzing activity of anthocyanins. Moreover, adding single or mixed of SLY-4E, F2-24E and HX-13E to wines could increase the content of varietal aroma and fermentative aroma compounds, especially terpenes, benzene derivatives, higher alcohols, and esters, which enhanced the fruity and floral flavor of wines. Hence, we expect that the potential application of crude extracts from the non-*Saccharomyces* yeasts in winemaking can provide an approach to improve flavor complexity and characteristics of wines. However, it is still unclear what effect is possessed between multiple β-glucosidases and the effect on volatile acids and aroma compounds. In the future, investigations on the mechanism of the synergistic effect of β-glucosidases on increasing aroma compound varieties and content might be required for further studies.

**Table 3 jof-08-00710-t003:** Volatile aroma compound concentration in wine treated by adding mixed yeast crude.extracts.

Compounds	Concentration (mg/L)								Threshold	OAV	Description
	SCw	S-Ew	H-Ew	F-Ew	S/H-Ew	S/F-Ew	H/F-Ew	S/H/F-Ew			
volatile aroma											
C6 compound											
1-Hexanol	0.76 ± 0.01 ^b^	0.66 ± 0.01 ^de^	0.63 ± 0.03 ^e^	0.71 ± 0.01 ^c^	0.69 ± 0.02 ^cd^	0.93 ± 0.01 ^a^	0.96 ± 0.03 ^a^	0.96 ± 0.07 ^a^	8 [32]	<0.1	Grass [44]
subtotal	0.76 ± 0.01 ^b^	0.66 ± 0.01 ^de^	0.63 ± 0.03 ^e^	0.71 ± 0.01 ^c^	0.69 ± 0.02 ^cd^	0.93 ± 0.01 ^a^	0.96 ± 0.03 ^a^	0.96 ± 0.07 ^a^			
Terpenes											
Linalool	0.21 ± 0.02 ^e^	0.30 ± 0.01 ^c^	0.31 ± 0.01 ^c^	0.28 ± 0.01 ^d^	0.29 ± 0.02 ^cd^	0.38 ± 0.01 ^a^	0.33 ± 0.00 ^b^	0.40 ± 0.05 ^a^	0.1 [25]	>1	Rose, fruit [45]
Citronellol	0.06 ± 0.01 ^d^	0.09 ± 0.01 ^bc^	0.09 ± 0.02 ^bc^	0.06 ± 0.01 ^d^	0.09 ± 0.02 ^bc^	0.10 ± 0.02 ^b^	0.07 ± 0.02 ^cd^	0.12 ± 0.01 ^a^	0.1 [32]	0.1–1	Lime [44]
1-Octen-3-ol	0.07 ± 0.01 ^e^	0.11 ± 0.00 ^d^	0.15 ± 0.02 ^c^	0.17 ± 0.01 ^b^	0.16 ± 0.01 ^bc^	0.16 ± 0.01 ^bc^	0.17 ± 0.00 ^b^	0.23 ± 0.01 ^a^	0.02 [28]	>1	Mushroom [46]
Geranyl acetone	Nd	0.06 ± 0.00 ^b^	0.02 ± 0.00 ^f^	0.03 ± 0.01 ^e^	0.04 ± 0.00 ^d^	0.07 ± 0.00 ^a^	0.05 ± 0.00 ^c^	0.04 ± 0.01 ^d^	0.06 [32]	0.1–1	Flower [44]
Nerolidol	0.03 ± 0.01 ^c^	0.04 ± 0.01 ^b^	0.05 ± 0.01 ^a^	0.03 ± 0.00 ^c^	0.04 ± 0.00 ^b^	0.05 ± 0.01 ^a^	0.05 ± 0.01 ^a^	0.05 ± 0.01 ^a^	0.7 [32]	<0.1	Rose, Apple, Orange [44]
Terpineol	Nd	Nd	0.02 ± 0.00 ^b^	Nd	0.01 ± 0.00 ^c^	Nd	0.02 ± 0.00 ^b^	0.03 ± 0.00 ^a^	0.25 [32]	<0.1	Flower, Pine [44]
Geraniol	Nd	Nd	0.01 ± 0.00 ^b^	0.01 ± 0.00 ^b^	Nd	0.01 ± 0.00 ^b^	0.01 ± 0.00 ^b^	0.02 ± 0.00 ^a^	0.03 [32]	<0.1	Rose, Geranium [47]
subtotal	0.37 ± 0.04 ^g^	0.60 ± 0.02 ^ef^	0.66 ± 0.06 ^d^	0.58 ± 0.04 ^f^	0.64 ± 0.05 ^de^	0.77 ± 0.04 ^b^	0.70 ± 0.04 ^c^	0.90 ± 0.09 ^a^			
total	1.14 ± 0.02 ^d^	1.26 ± 0.01 ^c^	1.29 ± 0.04 ^c^	1.29 ± 0.02 ^c^	1.34 ± 0.03 ^c^	1.70 ± 0.02 ^b^	1.66 ± 0.03 ^b^	1.86 ± 0.06 ^a^			
Fermentation aroma											
Benzene derivatives											
Benzaldehyde	0.20 ± 0.02 ^c^	0.38 ± 0.10 ^b^	0.35 ± 0.03 ^b^	0.36 ± 0.02 ^b^	0.38 ± 0.09 ^b^	0.41 ± 0.03 ^b^	0.39 ± 0.09 ^b^	0.52 ± 0.02 ^a^	2 [32]	0.1–1	Roasted almonds [44]
Benzyl alcohol	0.03 ± 0.00 ^d^	0.08 ± 0.01 ^b^	0.06 ± 0.01 ^c^	0.06 ± 0.00 ^c^	0.05 ± 0.01 ^c^	0.05 ± 0.00 ^c^	0.06 ± 0.02 ^c^	0.09 ± 0.01 ^a^	200 [25]	<0.1	Almond [45]
Phenylacetaldehyde	0.05 ± 0.00 ^d^	0.08 ± 0.01 ^bc^	0.12 ± 0.02 ^a^	0.07 ± 0.00 ^c^	0.11 ± 0.01 ^a^	0.09 ± 0.02 ^b^	0.07 ± 0.01 ^c^	0.09 ± 0.01 ^b^	0.005 [36]	>1	Flower, Rose, Honey [48]
Phenyl ethanol	50.76 ± 4.45 ^f^	110.76 ± 5.32 ^b^	102.60 ± 6.32 ^c^	85.37 ± 4.53 ^e^	105.43 ± 4.96 ^bc^	89.79 ± 5.10 ^de^	93.80 ± 3.78 ^d^	121.64 ± 3.54 ^a^	7.5 [25]	>1	Musk, Peach [45]
Phenethyl acetate	6.28 ± 0.33 ^d^	9.75 ± 0.54 ^b^	10.18 ± 1.05 ^b^	7.84 ± 0.85 ^c^	10.08 ± 0.93 ^b^	10.42 ± 0.59 ^b^	9.56 ± 0.39 ^b^	13.92 ± 0.35 ^a^	0.65 [25]	>1	Fruit, Flower [45]
subtotal	57.31 ± 4.82 ^f^	121.05 ± 5.98 ^b^	113.32 ± 7.42 ^c^	93.70 ± 5.41 ^e^	116.06 ± 6.01 ^c^	100.76 ± 5.75 ^d^	103.88 ± 4.30 ^d^	136.26 ± 3.94 ^a^			
Higher alcohols											
1-Butanol	0.75 ± 0.03 ^c^	0.81 ± 0.34 ^bc^	0.76 ± 0.01 ^c^	1.23 ± 0.09 ^a^	0.77 ± 0.04 ^bc^	0.79 ± 0.39 ^bc^	1.04 ± 0.14 ^ab^	1.02 ± 0.16 ^ab^	150 [25]	<0.1	Fragrant [45]
Isoamyl alcohol	90.98 ± 4.38 ^g^	129.74 ± 6.68 ^f^	230.17 ± 2.27 ^c^	251.61 ± 7.14 ^b^	230.11 ± 10.29 ^c^	196.84 ± 2.39 ^e^	219.07 ± 1.09 ^d^	267.10 ± 5.23 ^a^	30 [25]	>1	Bitter almond [45]
2-Methyl-1- butanol	42.76 ± 1.28 ^e^	51.69 ± 1.50 ^d^	56.20 ± 1.29 ^c^	61.16 ± 3.60 ^b^	52.61 ± 1.72 ^d^	64.23 ± 1.06 ^a^	40.58 ± 0.32 ^e^	65.55 ± 2.54 ^a^	65 [26]	0.1–1	
4-Methyl-1-pentanol	0.12 ± 0.02 ^c^	0.20 ± 0.05 ^bc^	0.36 ± 0.22 ^a^	0.20 ± 0.04 ^bc^	0.21 ± 0.01 ^bc^	0.32 ± 0.12 ^ab^	0.29 ± 0.02 ^ab^	0.38 ± 0.20 ^a^	50 [29]	<0.1	Almond [49]
3-Ethoxy-1-propanol	Nd	0.38 ± 0.01 ^b^	Nd	Nd	Nd	0.46 ± 0.00 ^a^	Nd	0.34 ± 0.03 ^c^			
3-Methyl-1-pentanol	0.54 ± 0.01 ^e^	0.81 ± 0.08 ^d^	1.04 ± 0.01 ^b^	1.03 ± 0.04 ^b^	0.86 ± 0.00 ^c^	1.05 ± 0.02 ^b^	1.21 ± 0.05 ^a^	1.24 ± 0.02 ^a^	0.5 [34]	>1	Earthy, Mushroom [50]
Heptanol	0.08 ± 0.00 ^e^	0.12 ± 0.00 ^d^	0.13 ± 0.01 ^c^	0.16 ± 0.00 ^b^	0.13 ± 0.00 ^c^	0.22 ± 0.02 ^a^	0.15 ± 0.00 ^b^	0.23 ± 0.01 ^a^	0.2–0.3 [34]	0.1–1	Lemon, Orange [50]
Isooctanol	0.03 ± 0.00 ^d^	0.03 ± 0.00 ^d^	0.04 ± 0.01 ^c^	0.03 ± 0.00 ^d^	0.04 ± 0.00 ^c^	0.06 ± 0.00 ^a^	0.04 ± 0.00 ^c^	0.05 ± 0.00 ^b^			Sweet, Flower, Rose [51]
1-Octanol	0.16 ± 0.01 ^f^	0.25 ± 0.01 ^de^	0.25 ± 0.04 ^de^	0.27 ± 0.01 ^cd^	0.23 ± 0.00 ^e^	0.31 ± 0.00 ^a^	0.28 ± 0.00 ^bc^	0.30 ± 0.01 ^ab^	0.9 [25]	0.1–1	Orange, Vanilla [45]
1-Nonanol	0.06 ± 0.00 ^d^	0.12 ± 0.03 ^c^	0.13 ± 0.01 ^c^	0.15 ± 0.02 ^b^	0.15 ± 0.00 ^b^	0.22 ± 0.01 ^a^	0.14 ± 0.00 ^bc^	0.15 ± 0.00 ^b^	0.015 [25]	>1	Orange [45]
Decanol	0.03 ± 0.00 ^e^	0.07 ± 0.01 ^b^	0.07 ± 0.00 ^b^	0.07 ± 0.01 ^b^	0.05 ± 0.01 ^d^	0.06 ± 0.01 ^c^	0.06 ± 0.00 ^c^	0.08 ± 0.00 ^a^	0.4 [25]	0.1–1	Flower [45]
subtotal	135.51 ± 5.74 ^f^	184.22 ± 8.71 ^e^	289.15 ± 3.86 ^c^	315.89 ± 10.96 ^b^	285.17 ± 12.08 ^c^	264.56 ± 4.02 ^d^	262.86 ± 1.63 ^d^	336.42 ± 8.21 ^a^			
Acetate esters											
Isobutyl acetate	0.03 ± 0.00 ^d^	0.03 ± 0.00 ^d^	0.02 ± 0.00 ^e^	0.09 ± 0.00 ^a^	0.04 ± 0.00 ^c^	0.05 ± 0.00 ^b^	Nd	Nd	1.6 [32]	<0.1	Banana [52]
Isoamyl acetate	4.39 ± 0.78 ^bc^	3.45 ± 0.53 ^d^	3.85 ± 0.03 ^cd^	8.01 ± 0.43 ^a^	4.03 ± 0.27 ^c^	4.82 ± 0.53 ^b^	4.16 ± 0.36 ^c^	1.66 ± 0.03 ^e^	0.2 [25]	>1	Banana [45], Green apple [46]
2-Methylbutyl acetate	0.42 ± 0.07 ^c^	0.36 ± 0.01 ^d^	0.42 ± 0.01 ^c^	0.76 ± 0.04 ^a^	0.44 ± 0.04 ^c^	0.50 ± 0.02 ^b^	0.42 ± 0.02 ^c^	0.16 ± 0.01 ^e^	0.16 [26]	>1	
Hexyl acetate	0.06 ± 0.00 ^bc^	0.04 ± 0.01 ^d^	0.07 ± 0.00 ^b^	0.08 ± 0.00 ^a^	0.04 ± 0.01 ^d^	0.06 ± 0.01 ^b^	0.05 ± 0.01 ^c^	0.04 ± 0.00 ^d^	1.5 [32]	<0.1	Fruit, Pear, Cherry [44]
subtotal	4.90 ± 0.85 ^c^	3.87 ± 0.56 ^e^	4.36 ± 0.04 ^d^	8.94 ± 0.47 ^a^	4.55 ± 0.32 ^cd^	5.43 ± 0.55 ^b^	4.64 ± 0.39 ^cd^	1.86 ± 0.04 ^f^			
Fatty acid ethyl esters											
Ethyl propionate	0.11 ± 0.01 ^d^	0.14 ± 0.00 ^c^	0.11 ± 0.00 ^d^	0.20 ± 0.00 ^a^	0.15 ± 0.01 ^b^	0.16 ± 0.01 ^b^	0.10 ± 0.00 ^e^	0.05 ± 0.00 ^f^	1.8 [32]	<0.1	Pineapple, Banana, Apple [46]
Ethyl butyrate	0.20 ± 0.01 ^bcd^	0.18 ± 0.01 ^d^	0.19 ± 0.01 ^cd^	0.41 ± 0.05 ^a^	0.20 ± 0.01 ^bc^	0.22 ± 0.00 ^b^	0.21 ± 0.01 ^bc^	0.08 ± 0.00 ^e^	0.02 [32]	>1	Strawberry, Apple, Banana [44]
Ethyl hexanoate	4.00 ± 0.14 ^de^	3.72 ± 0.28 ^e^	4.60 ± 0.12 ^bc^	5.25 ± 0.60 ^a^	4.33 ± 0.29 ^bcd^	4.65 ± 0.07 ^b^	4.25 ± 0.05 ^cd^	3.27 ± 0.40 ^f^	0.014 [25]	>1	Green apple, Fennel [45]
Ethyl heptanoate	0.04 ± 0.01 ^b^	0.04 ± 0.00 ^b^	0.06 ± 0.00 ^a^	0.04 ± 0.00 ^b^	0.04 ± 0.01 ^b^	0.04 ± 0.00 ^b^	0.03 ± 0.00 ^c^	0.03 ± 0.00 ^c^	0.002 [28]	>1	Sweet, Strawberry, Banana [53]
Ethyl octanoate	9.65 ± 0.41 ^e^	10.93 ± 0.59 ^d^	14.2 ± 0.26 ^b^	17.35 ± 0.80 ^a^	16.92 ± 1.17 ^a^	17.18 ± 0.79 ^a^	12.57 ± 0.34 ^c^	13.16 ± 0.84 ^c^	0.25 [25]	>1	Fruit [45]
Ethyl nonanoate	0.09 ± 0.00 ^g^	0.13 ± 0.00 ^f^	0.15 ± 0.00 ^e^	0.25 ± 0.00 ^a^	0.22 ± 0.00 ^b^	0.24 ± 0.01 ^a^	0.20 ± 0.01 ^c^	0.18 ± 0.01 ^d^	1.3 [32]	0.1–1	Flower, Fruit [52]
Ethyl sebacate	0.49 ± 0.00 ^e^	0.72 ± 0.07 ^d^	0.95 ± 0.00 ^c^	0.34 ± 0.00 ^f^	1.17 ± 0.07 ^b^	1.35 ± 0.05 ^a^	0.38 ± 0.05 ^f^	1.19 ± 0.08 ^b^			
Ethyl decanoate	5.10 ± 0.52 ^f^	7.33 ± 0.54 ^e^	9.13 ± 0.46 ^d^	12.32 ± 0.66 ^b^	12.81 ± 0.42 ^b^	12.61 ± 0.19 ^b^	10.29 ± 0.25 ^c^	13.55 ± 0.54 ^a^	0.2 [25]	>1	Apple, Flower [45]
Ethyl laurate	0.87 ± 0.02 ^g^	1.61 ± 0.00 ^f^	1.83 ± 0.01 ^e^	1.98 ± 0.02 ^d^	2.10 ± 0.02 ^c^	2.13 ± 0.06 ^c^	2.57 ± 0.04 ^b^	2.75 ± 0.06 ^a^	1.5 [29]	>1	Fruit, Fatty [49]
Ethyl tetradecanoate	0.07 ± 0.01 ^e^	0.09 ± 0.01 ^d^	0.10 ± 0.00 ^d^	0.12 ± 0.00 ^c^	0.13 ± 0.00 ^b^	0.11 ± 0.00 ^c^	0.15 ± 0.00 ^a^	0.16 ± 0.00 ^a^	2 [34]	<0.1	Sweet fruit, Butter, Fatty [50]
Hexadecanoic acid, ethyl ester	0.16 ± 0.00 ^f^	0.22 ± 0.00 ^e^	0.22 ± 0.00 ^e^	0.24 ± 0.00 ^d^	0.30 ± 0.01 ^c^	0.31 ± 0.00 ^b^	0.31 ± 0.00 ^b^	0.38 ± 0.01 ^a^	1.5 [30]	0.1–1	Fruit, Sweet, Fatty [51]
Diethyl succinate	0.32 ± 0.01 ^f^	0.57 ± 0.00 ^bc^	0.54 ± 0.06 ^c^	0.40 ± 0.01 ^e^	0.59 ± 0.05 ^b^	0.54 ± 0.02 ^c^	0.44 ± 0.02 ^d^	0.67 ± 0.02 ^a^	200 [25]	<0.1	Fruit, Melon [52]
subtotal	21.08 ± 1.12 ^e^	25.66 ± 1.50 ^d^	32.07 ± 0.93 ^c^	38.89 ± 2.15 ^a^	38.96 ± 2.07 ^a^	39.55 ± 1.20 ^a^	31.50 ± 0.78 ^c^	35.46 ± 1.96 ^b^			
Other esters											
Hexanoic acid, 3-methylbutyl ester	0.04 ± 0.00 ^e^	0.05 ± 0.00 ^d^	0.06 ± 0.00 ^c^	0.09 ± 0.01 ^a^	0.08 ± 0.00 ^b^	0.08 ± 0.00 ^b^	0.05 ± 0.00 ^d^	0.06 ± 0.00 ^c^	1 [35]	<0.1	Apple, Pineapple [49]
Hexanoic acid, 2-methylbutyl ester	Nd	0.03 ± 0.00 ^b^	0.02 ± 0.00 ^c^	0.03 ± 0.00 ^b^	0.03 ± 0.00 ^b^	0.04 ± 0.00 ^a^	0.02 ± 0.00 ^c^	0.03 ± 0.00 ^b^			
Isoamyl octanoate	0.09 ± 0.01 ^f^	0.14 ± 0.00 ^e^	0.16 ± 0.01 ^d^	0.23 ± 0.01 ^a^	0.21 ± 0.01 ^c^	0.22 ± 0.01 ^b^	0.21 ± 0.01 ^c^	0.24 ± 0.00 ^a^	0.125 [29]	>1	Fruit, Cheese [49]
Octanoic acid, 2-methylbutyl ester	0.03 ± 0.01 ^e^	0.08 ± 0.01 ^d^	0.09 ± 0.00 ^d^	0.12 ± 0.01 ^c^	0.11 ± 0.01 ^c^	0.11 ± 0.00 ^c^	0.13 ± 0.00 ^b^	0.15 ± 0.00 ^a^			
subtotal	0.17 ± 0.01 ^d^	0.30 ± 0.01 ^c^	0.32 ± 0.02 ^c^	0.48 ± 0.02 ^a^	0.43 ± 0.02 ^b^	0.45 ± 0.02 ^b^	0.41 ± 0.01 ^b^	0.47 ± 0.01 ^a^			
total esters	26.15 ± 1.99 ^e^	29.84 ± 2.07 ^d^	36.75 ± 0.98 ^c^	48.31 ± 2.65 ^a^	43.95 ± 2.40 ^b^	45.43 ± 1.77 ^b^	36.55 ± 1.18 ^c^	37.80 ± 2.01 ^c^			
Fatty acids											
Isobutyric acid	Nd	0.02 ± 0.00 ^a^	0.01 ± 0.00 ^b^	Nd	0.01 ± 0.00 ^b^	Nd	Nd	0.02 ± 0.00 ^a^	2.3 [26]	<0.1	Sour, Cheese [49]
2-Methylbutyric acid	0.65 ± 0.04 ^b^	0.60 ± 0.06 ^c^	0.62 ± 0.01 ^bc^	0.55 ± 0.00 ^d^	0.44 ± 0.06 ^e^	0.37 ± 0.01 ^f^	0.59 ± 0.00 ^cd^	0.78 ± 0.02 ^a^	0.05 [25]	>1	
Isopentanoic acid	0.33 ± 0.02 ^ab^	0.38 ± 0.01 ^a^	0.36 ± 0.03 ^a^	0.27 ± 0.04 ^c^	0.27 ± 0.06 ^c^	0.26 ± 0.04 ^c^	0.31 ± 0.06 ^bc^	0.27 ± 0.02 ^c^	3 [29]	<0.1	Fatty [49]
Octanoic acid	0.92 ± 0.03 ^e^	1.06 ± 0.05 ^d^	1.13 ± 0.00 ^c^	1.16 ± 0.01 ^bc^	1.17 ± 0.02 ^bc^	1.19 ± 0.10 ^bc^	1.21 ± 0.06 ^b^	1.52 ± 0.03 ^a^	15 [25]	<0.1	Sour, Cheese [45]
Decanoic acid	0.13 ± 0.00 ^g^	0.15 ± 0.00 ^f^	0.19 ± 0.00 ^e^	0.22 ± 0.00 ^c^	0.21 ± 0.00 ^d^	0.23 ± 0.00 ^b^	0.19 ± 0.00 ^e^	0.26 ± 0.01 ^a^	8 [25]	<0.1	An unpleasant fatty [45]
subtotal	2.03 ± 0.09 ^d^	2.21 ± 0.12 ^bc^	2.31 ± 0.05 ^b^	2.20 ± 0.05 ^c^	2.10 ± 0.15 ^d^	2.05 ± 0.16 ^d^	2.31 ± 0.13 ^b^	2.85 ± 0.08 ^a^			
Carbonyl compounds											
3-Methyl-butanal	Nd	Nd	Nd	0.01 ± 0.00 ^a^	0.01 ± 0.00 ^a^	Nd	0.01 ± 0.00 ^b^	Nd			
2,3-Pentanedione	0.20 ± 0.01 ^h^	0.33 ± 0.00 ^b^	0.3 ± 0.00 ^c^	0.35 ± 0.01 ^a^	0.24 ± 0.00 ^g^	0.26 ± 0.00 ^e^	0.25 ± 0.00 ^f^	0.28 ± 0.00 ^d^	2 [35]	0.1–1	Pecan [46]
Nonanal	0.04 ± 0.00 ^c^	0.05 ± 0.00 ^b^	0.06 ± 0.00 ^a^	0.04 ± 0.00 ^c^	0.04 ± 0.00 ^c^	0.05 ± 0.00 ^b^	0.04 ± 0.00 ^c^	0.04 ± 0.00 ^c^	0.015 [32]	>1	Herb, Slightly spicy [44]
Decanal	0.05 ± 0.00 ^c^	0.08 ± 0.01 ^a^	0.06 ± 0.00 ^b^	0.06 ± 0.00 ^b^	0.04 ± 0.00 ^d^	0.06 ± 0.01 ^b^	0.05 ± 0.00 ^c^	0.06 ± 0.01 ^b^	0.001 [30]	>1	Flower [51]
2,4-Dimethoxybenzaldehyde	0.36 ± 0.02 ^e^	0.43 ± 0.03 ^d^	0.46 ± 0.05 ^cd^	0.72 ± 0.03 ^a^	0.53 ± 0.01 ^b^	0.49 ± 0.05 ^c^	0.25 ± 0.01 ^f^	0.43 ± 0.01 ^d^			
subtotal	0.65 ± 0.03 ^e^	0.89 ± 0.04 ^b^	0.87 ± 0.05 ^bc^	1.18 ± 0.04 ^a^	0.87 ± 0.02 ^bc^	0.85 ± 0.06 ^c^	0.59 ± 0.02 ^f^	0.81 ± 0.02 ^d^			
total	221.66 ± 5.17 ^f^	338.21 ± 6.91 ^e^	442.40 ± 5.05 ^c^	461.30 ± 7.80 ^b^	448.15 ± 8.44 ^c^	413.65 ± 4.80 ^d^	406.20 ± 2.96 ^d^	514.14 ± 5.82 ^a^			

Note: SPSS software was used for ANOVA analysis, and the different letters in the same row show significant difference (*p* < 0.05). ND means no substance was detected.

## Figures and Tables

**Figure 1 jof-08-00710-f001:**
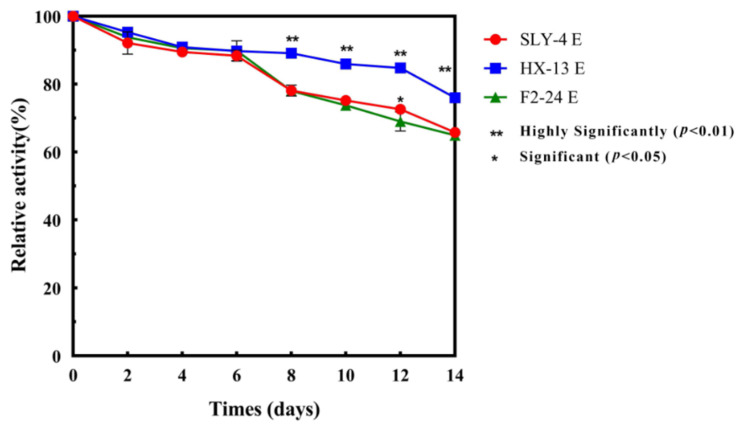
Stability of crude extracts of *β*-glucosidase in modified P-C buffer.

**Figure 2 jof-08-00710-f002:**
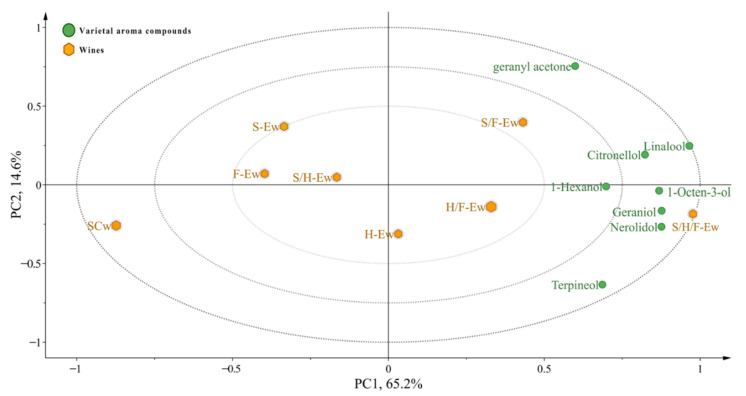
PCA of varietal aroma compounds from wines.

**Figure 3 jof-08-00710-f003:**
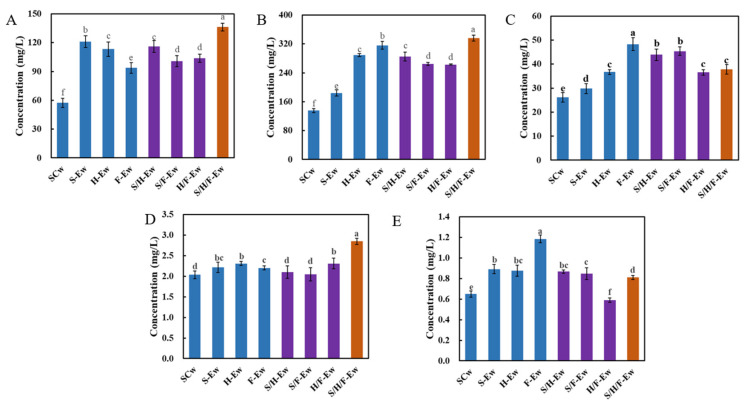
Fermentative aroma compound content in wines. (**A**) Benzene derivatives; (**B**) higher alcohols; (**C**) esters; (**D**) fatty acids; (**E**) carbonyl compounds. Different letters on the histogram showed significant difference (*p* < 0.05).

**Figure 4 jof-08-00710-f004:**
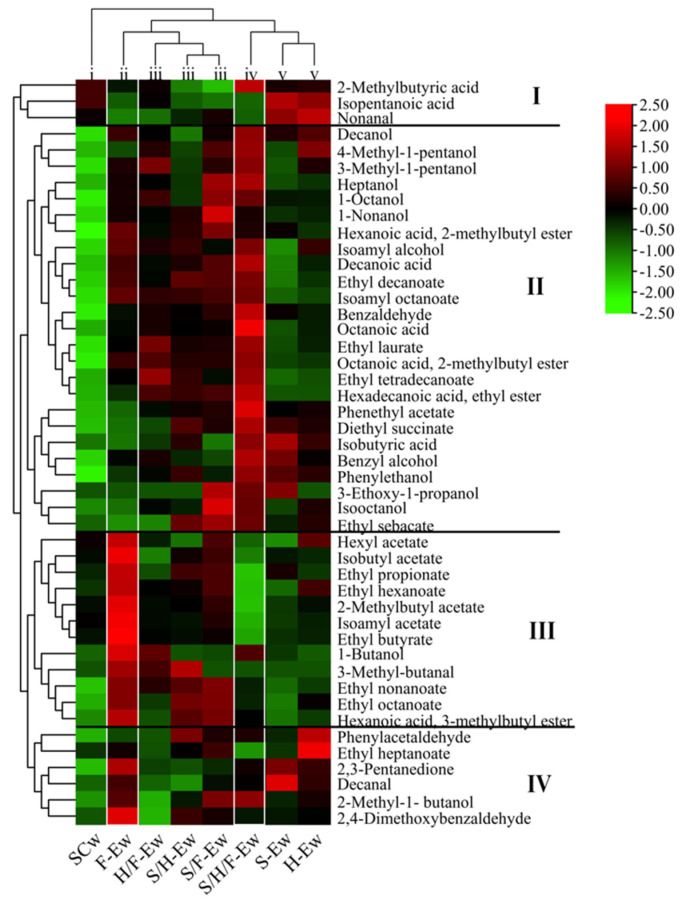
Hierarchical cluster analysis of aroma compounds in wines. All the detected fermentative aroma compounds were clustered into four classes and designated as class I, II, III, and IV.

**Figure 5 jof-08-00710-f005:**
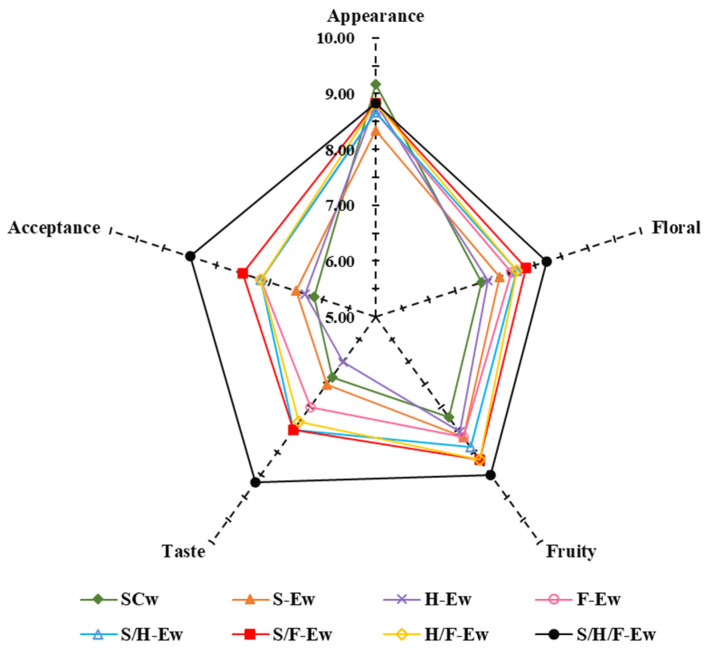
Sensory evaluation of wines.

**Table 1 jof-08-00710-t001:** Adding strategies of crude extracts of β-glucosidase in wines.

Wines	Add the Crude Extracts of β-Glucosidase From		
	SLY-4E	HX-13E	F2-24E
SCw			
S-Ew	yes		
H-Ew		yes	
F-Ew			yes
S/H-Ew	yes	yes	
S/F-Ew	yes		yes
H/F-Ew		yes	yes
S/H/F-Ew	yes	yes	yes

**Table 2 jof-08-00710-t002:** Physicochemical characteristics of wines.

Wines	Alcohol (%, *v*/*v*)	Total Acids (g/L)	Volatile Acids (g/L)	Anthocyanin (mg/L)
SCw	12.56 ± 0.53 ^a^	5.81 ± 0.22 ^a^	0.68 ± 0.00 ^a^	187.79 ± 1.25 ^a^
S-Ew	12.07 ± 0.14 ^b^	5.81 ± 0.22 ^a^	0.61 ± 0.01 ^c^	182.83 ± 0.42 ^c^
H-Ew	12.13 ± 0.17 ^ab^	5.53 ± 0.19 ^ab^	0.68 ± 0.04 ^a^	163.30 ± 0.84 ^f^
F-Ew	12.21 ± 0.13 ^ab^	5.20 ± 0.32 ^b^	0.49 ± 0.02 ^d^	176.08 ± 0.84 ^d^
S/H-Ew	12.26 ± 0.23 ^ab^	5.39 ± 0.28 ^ab^	0.61 ± 0.04 ^c^	173.24 ± 0.05 ^e^
S/F-Ew	12.51 ± 0.11 ^ab^	5.44 ± 0.22 ^ab^	0.65 ± 0.07 ^b^	184.13 ± 2.93 ^bc^
H/F-Ew	12.25 ± 0.23 ^ab^	5.20 ± 0.49 ^b^	0.49 ± 0.01 ^d^	172.17 ± 3.76 ^e^
S/H/F-Ew	12.33 ± 0.09 ^ab^	5.72 ± 0.19 ^a^	0.61 ± 0.03 ^c^	184.96 ± 2.09 ^b^

Note: different letters in the same column showed significant difference (*p* < 0.05).

## Data Availability

Not applicable.

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
