# Peer review of "Effects of Mixed Adding Crude Extracts of β-Glucosidases from Three Different Non-*Saccharomyces* Yeast Strains on the Quality of Cabernet Sauvignon Wines"

_jof, 2022, doi:10.3390/jof8070710_

Round 1
Reviewer 1 Report
Title: Effects of mixed adding crude extracts of β-glucosidases from three different non-saccharomyces yeast strains on the quality of cabernet sauvignon wines.
Authors: Jing Liao, Shuangmei Zhang and Xiuyan Zhang
Summary: The study investigated the effects of crude extracts of β-glucosidase from Issatchenkia terricola (SLY-4), Pichia kudriavzevii (F2-24) and Metschnikowia pulcherrima (HX-13) on the flavor complexity and typicality of Cabernet Sauvignon wines in comparison to Saccharomyces cerevisiae. The study compares the effect of individual enzyme against the crude extracts. The three enzymes attenuated the amount of anthocyanin, total acids and volatile acids in wine, however, they concomitantly increased terpenes, benzene derivatives, higher alcohols and esters. A mixture of two or more enzymes produced contrasting effects, whereas combination of the three enzymes produced the best wine flavor. The study gives insight into non-Saccharomyces yeast options suitable for wine making as well as for enhancing wine flavor. Despite its relevance, there are few issues to be reviewed as well as minor corrections to be done in the manuscript as indicated in the comments below:
Comments:
1. The phrase “mixed adding crude extracts” in the title can be edited to “crude extract mixtures” for better understanding
2. The phrase “mixed adding” in the entire manuscript can be changed to “mix/mixture/blend”
3. Line 40 and 41 is a single sentence, use comma instead full stop, or rephrase the sentence.
4. The sentence beginning at line pauses with “respectively” which doesn’t relate to anything in the preceding sentence
5. “respectively” at the end of sentence in line 65 must be after ammonium sulfate in line 64
6. “respectively” at the end of sentence in line 80 must be removed.
7. Fig 1 and Table 2: It was concluded that the addition of the crude extract and purified β-glucosidase from yeasts would decrease anthocyanin content. How is this assertion reflected in the table when some crude mix extracts have higher anthocyanin content than the single enzyme?
8. Fig 2. How was the grouping performed? Why is terpineol not included in grouping? The three enzymes showed slightly higher average 1-hexanol and linalool but the standard deviation widens the data spread as compared to S/F-Ew. Why are these 2 compounds associated with only S/H/F-Ew and not S/F-Ew despite having same significance in table 3?
9. The sentence “H/F-Ew, S/H-Ew and S/F-Ew were richer in and S/H/F-Ew” in lines 221 and 222 seems to have something missing
10. Rephrase this sentence in lines 230 to 232: “It illustrated that mixed adding SLY-4E, F2-24E and HX-13E affected the formation of fermentative aroma compounds multiply”.
11. Fig 4. What was the reference/control to explain the expression range of -2.5 to 2.5? Is this range fold-change or an average of released compounds? What does 0 on this scale imply?
12. Aside class I which gives herb flavor, what is the essence of the remaining classes? How does each class contribute to quality of wine? Which compounds are most important in each class and how do they affect wine quality?
13. Edit this phrase in line 248: “The promote of wine taste”.
14. Figure 5: How does volatile acidity explains the taste score in this plot? What factors contributed to the improved taste by S/H/F-Ew?
Author Response
Dear Reviewers,
Thank you very much for your time involved in reviewing the manuscript and your very encouraging comments on the merits.
Comments:
The study investigated the effects of crude extracts of β-glucosidase from Issatchenkia terricola (SLY-4), Pichia kudriavzevii (F2-24) and Metschnikowia pulcherrima (HX-13) on the flavor complexity and typicality of Cabernet Sauvignon wines in comparison to Saccharomyces cerevisiae. The study compares the effect of individual enzyme against the crude extracts. The three enzymes attenuated the amount of anthocyanin, total acids and volatile acids in wine, however, they concomitantly increased terpenes, benzene derivatives, higher alcohols and esters. A mixture of two or more enzymes produced contrasting effects, whereas combination of the three enzymes produced the best wine flavor. The study gives insight into non-Saccharomyces yeast options suitable for wine making as well as for enhancing wine flavor.
We also appreciate your clear and detailed feedback and hope that the explanation has fully addressed all of your concerns. In the remainder of this letter, we discuss each of your comments individually along with our corresponding responses.
Comment 1:
The phrase “mixed adding crude extracts” in the title can be edited to “crude extract mixtures” for better understanding.
Response 1:
Thank you for the detailed review. We have carefully and thoroughly proofread the manuscript to correct all the phrases.
Comment 2:
The phrase “mixed adding” in the entire manuscript can be changed to “mix/mixture/blend”
Response 2:
Thank you for the detailed review. We have carefully and thoroughly proofread the manuscript to correct all the phrases.
Comment 3:
Line 40 and 41 is a single sentence, use comma instead full stop, or rephrase the sentence.
Response 3:
Thank you for the detailed review. We have rephrased the sentence in line 40 and 41.
Comment 4:
The sentence beginning at line pauses with “respectively” which doesn’t relate to anything in the preceding sentence.
Response 4:
Thank you for the detailed review. We have deleted the word “respectively”.
Comment 5:
“respectively” at the end of sentence in line 65 must be after ammonium sulfate in line 64
Response 5:
Thank you for the detailed review. We have deleted the word “respectively” in line 64.
Comment 6:
“respectively” at the end of sentence in line 80 must be removed.
Response 6:
Thank you for the detailed review. We have deleted the word “respectively” in line 80.
Comment 7:
Fig 1 and Table 2: It was concluded that the addition of the crude extract and purified β-glucosidase from yeasts would decrease anthocyanin content. How is this assertion reflected in the table when some crude mix extracts have higher anthocyanin content than the single enzyme?
Response 7:
Thank you for the detailed review. In table 2, it shows that the anthocyanin content of H-Ew was 163.30 mg/L which was lower than S-Ew, F-Ew. The anthocyanin content of S/H-Ew (173.08 mg/L) and H/F-Ew (172.17 mg/L) and S/H/F-Ew (184.96 mg/L) were higher H-Ew, so that we supposed that the combinational addition of SLY-4E, F2-24E and HX-13E could attenuate the declining trend. Besides, we were not describing the data in enough detail and we have carefully and thoroughly added the relevant description in more detail.
Comment 8:
Fig 2. How was the grouping performed? Why is terpineol not included in grouping? The three enzymes showed slightly higher average 1-hexanol and linalool but the standard deviation widens the data spread as compared to S/F-Ew. Why are these 2 compounds associated with only S/H/F-Ew and not S/F-Ew despite having same significance in table 3?
Response 8:
Thank you for the detailed review. Figure 2 does have an error and we have carefully and thoroughly proofread the manuscript to correct it.
Comment 9:
The sentence “H/F-Ew, S/H-Ew and S/F-Ew were richer in and S/H/F-Ew” in lines 221 and 222 seems to have something missing
Response 9:
Thank you for the detailed review. We have corrected the sentence in line 221.
Comment 10:
Rephrase this sentence in lines 230 to 232: “It illustrated that mixed adding SLY-4E, F2-24E and HX-13E affected the formation of fermentative aroma compounds multiply”.
Response 10:
Thank you for the detailed review. We have rephrased this sentence in lines 230 to 232.
Comment 11:
Fig 4. What was the reference/control to explain the expression range of -2.5 to 2.5? Is this range fold-change or an average of released compounds? What does 0 on this scale imply?
Response 11:
Thank you for the detailed review. Heatmap visualization analysis of the wines and their flavor compounds were performed after the Z-score standardization. Descriptions on the handling of the data are not sufficiently detailed and we have been corrected carefully.
Comment 12:
Aside class I which gives herb flavor, what is the essence of the remaining classes? How does each class contribute to quality of wine? Which compounds are most important in each class and how do they affect wine quality?
Response 12:
Thank you for the detailed review. Both classes II and III contained various compounds, and most of them having floral and fruity notes. The contribution of each type of aroma compound to the wine was analyzed through the OVA (odor active value) which was showed in Table.3. The results showed that in class II, the OVA of phenyl ethanol, phenethyl acetate, isoamyl alcohol, 3-methyl-1-pentanol, 1-nonanol, ethyl decanoate, ethyl laurate and isoamyl octanoate were greater than 1. These compounds presented a fruity, fatty, floral and earthy note and improved the complexity of aroma in S/H/F-Ew. In Class III, the OVA of isoamyl acetate, ethyl butyrate, ethyl hexanoate and ethyl octanoate was greater than one which improved fruity aroma in wines. Discussion on these results are not sufficiently detailed and we have been corrected carefully.
Comment 13:
Edit this phrase in line 248: “The promote of wine taste”.
Response 13:
Thank you for the detailed review. We have corrected the sentence in line 248.
Comment 14:
Figure 5: How does volatile acidity explains the taste score in this plot? What factors contributed to the improved taste by S/H/F-Ew?
Response 14:
Thank you for the detailed review. Major taste qualities in wine are sweetness, sourness and bitterness, contributed by sugars, organic acids and ethanol, respectively, while mouthfeel encompasses a number of inter-related tactile sensations. Acetic acid is mostly responsible for the sour and vinegary smell and taste in wines. We analyzed the volatile acid of wines which were mainly contained acetic acid. Therefore, we concluded that the decrease of volatile acid improved wine taste. Wine taste is influenced by many factors, but in this article, the analysis is not enough and should be focused on in the further work.Discussion on these results are not sufficiently detailed and we have been corrected carefully.
Reviewer 2 Report
A very interesting publication, covering issues related to the formation of compounds responsible for the sensorics of Cabernet Sauvignon Wines during the fermentation professor using a mixture of yeast, which has the ability to create the enzymes β-Glucosidases. I find this an interesting approach to the problem and its solution using natural resources. The research scope of the work has been well developed. The included research methods do not raise any objections. They were properly described. The research results are presented clearly in the form of charts and tables. The results were also statistically processed, which significantly facilitates the comparison and interpretation of the results. The presented conclusions are adequate to the obtained results. The cited literature is corrected used. It is worth emphasizing that it was correctly selected and developed. I suggest accepting it for printing.
Author Response
Dear Reviewers,
Thank you very much for your time involved in reviewing the manuscript and your very encouraging comments on the merits.
Comments:
A very interesting publication, covering issues related to the formation of compounds responsible for the sensorics of Cabernet Sauvignon Wines during the fermentation professor using a mixture of yeast, which has the ability to create the enzymes β-Glucosidases. I find this an interesting approach to the problem and its solution using natural resources. The research scope of the work has been well developed. The included research methods do not raise any objections. They were properly described. The research results are presented clearly in the form of charts and tables. The results were also statistically processed, which significantly facilitates the comparison and interpretation of the results. The presented conclusions are adequate to the obtained results. The cited literature is corrected used. It is worth emphasizing that it was correctly selected and developed. I suggest accepting it for printing.
Response:
We have carefully and thoroughly proofread the manuscript to correct all the grammar and typos.
Reviewer 3 Report
1)What is the novelty of the study?
2) Please, give more information about the analyzed wine.
3) Table 1 - the symbols of wine are illegible.
4) Poor number of analyzed wine samples.
5) Lines 94-95 - add more information/details about the methods of analyses.
6) Add more discussion in points 3.2 and 3.3
7) English correction.
Author Response
Dear Reviewers,
Thank you very much for your time involved in reviewing the manuscript and your very encouraging comments on the merits.
We also appreciate your clear and detailed feedback and hope that the explanation has fully addressed all of your concerns. In the remainder of this letter, we discuss each of your comments individually along with our corresponding responses.
Comment 1:
What is the novelty of the study?
Response 1:
β-glucosidases could hydrolyze non-volatile flavor precursors and generate aroma substances with volatile flavor. Issatchenkia terricola SLY-4, Pichia kudriavzevii F2-24 and Metschnikowia pulcherrima HX-13 with β-glucosidases activities or their crude extracts of β-glucosidase (named as SLY-4E, F2-24E and HX-13E) could improve the flavor complexity and typicality of wines and present different aroma compounds profiles by co-fermented with S. cerevisiae. However, their crude extracts of β-glucosidase on the flavor complexity and typicality of Cabernet Sauvignon wines were not determined. Besides, it is the first report about adding a combination of crude extracts of β-Glucosidases from three different non-Saccharomyces Yeast Strains on the quality of Cabernet Sauvignon wines.
Comment 2:
Please, give more information about the analyzed wine.
Response 2:
Thank you for the detailed review. The wine used in this paper was Cabernet Sauvignon Wines which were harvested in Helan Mountain East Foothill. After selection, destem and pressed, we got the grape must which contained 237.1 g/L sugar calculated as glucose and 5.3 g/L acid calculated as tartaric acid.
Comment 3:
Table 1 - the symbols of wine are illegible.
Response 3:
Thank you for the detailed review. Table 1 does have an error and we have carefully and thoroughly proofread the manuscript to correct it.
The relevant contents are provided below as a screen dump for your quick reference.
Comment 4:
Poor number of analyzed wine samples.
Response 4:
Thank you for the detailed review. This paper examines the adding a combination of crude extracts of β-glucosidases from Issatchenkia terricola SLY-4, Pichia kudriavzevii F2-24 and Metschnikowia pulcherrima HX-13 on the quality of Cabernet Sauvignon wines. A total of 9 wine samples were determined and there were no more permutation combinations.
Comment 5:
Lines 94-95 - add more information/details about the methods of analyses.
Response 5:
Thanks for your great suggestion on improving the accessibility of our manuscript. We have added more details about the methods of analyses.
Comment 6:
Add more discussion in points 3.2 and 3.3
Response 6:
Thanks for your great suggestion on improving the accessibility of our manuscript. We have added more discussion in points 3.2 and 3.3.
Comment 7:
English correction.
Response 7:
Thanks for your great suggestion on our manuscript. We have carefully and thoroughly proofread the manuscript to correct all the grammar and typos.
Special thanks to you for your good comments.
We appreciate for Reviewers’ warm work earnestly, and hope that the correction will meet with approval.
Once again, thank you very much for your comments and suggestions.
Reviewer 4 Report
Please see uploaded Word document

Author Response
Dear Reviewers,
Thank you very much for your time involved in reviewing the manuscript and your very encouraging comments on the merits. We also appreciate your clear and detailed feedback and hope that the explanation has fully addressed all of your concerns.
We have carefully and thoroughly proofread the manuscript to correct all every suggestion you addressed in the writing and content.
Special thanks to you for your good comments.
We appreciate for your warm work earnestly, and hope that the correction will meet with approval.
Once again, thank you very much for your comments and suggestions.
Round 2
Reviewer 3 Report
The manuscript titled Effects of Mixed Adding Crude Extracts of β-Glucosidases from Three Different Non-Saccharomyces Yeast Strains on the Quality of Cabernet Sauvignon Wines could be published in JoF.